# AISMPred: A Machine Learning Approach for Predicting Anti-Inflammatory Small Molecules

**DOI:** 10.3390/ph17121693

**Published:** 2024-12-15

**Authors:** Subathra Selvam, Priya Dharshini Balaji, Honglae Sohn, Thirumurthy Madhavan

**Affiliations:** 1Computational Biology Laboratory, Department of Genetic Engineering, School of Bioengineering, SRM Institute of Science and Technology, Kattankulathur, Chengalpattu 603203, Tamil Nadu, India; subathraselvam15@gmail.com (S.S.); priyamedigen@gmail.com (P.D.B.); 2Department of Chemistry, Chosun University, Gwangju 501-759, Republic of Korea

**Keywords:** anti-inflammatory, autoimmune disease, small molecules, machine learning, k-fold cross-validation

## Abstract

**Background/Objectives:** Inflammation serves as a vital response to diverse harmful stimuli like infections, toxins, or tissue injuries, aiding in the elimination of pathogens and tissue repair. However, persistent inflammation can lead to chronic diseases. Peptide therapeutics have gained attention for their specificity in targeting cells, yet their development remains costly and time-consuming. Therefore, small molecules, with their stability, low immunogenicity, and oral bioavailability, have become a focal point for predicting anti-inflammatory small molecules (AISMs). **Methods:** In this study, we introduce a computational method called AISMPred, designed to classify AISMs and non-AISMs. To develop this approach, we constructed a dataset comprising 1750 AISMs and non-AISMs, each annotated with ${IC}_{50}$ values sourced from the PubChem BioAssay database. We computed two distinct types of molecular descriptors using PaDEL and Mordred tools. Subsequently, these descriptors were concatenated to form a hybrid feature set. The SVC-L1 regularization method was implemented for the optimum feature selection to develop robust Machine learning (ML) models. Five different conventional ML classifiers were employed, such as RF, ET, KNN, LR, and Ensemble methods. **Results:** A total of 15 ML models were developed using 2D, FP, and Hybrid feature sets, with the ET model with hybrid features achieving the highest accuracy of 92% and an AUC of 0.97 on the independent test dataset. **Conclusions:** This study provides an effective method for screening AISMs, potentially impacting drug discovery and design.

## 1. Introduction

Inflammatory mechanisms are crucial components of the immune response, aiding the body’s defense against infections or injuries and maintaining tissue homeostasis under adverse conditions [1]. Inflammation is typically classified into two main types: acute and chronic. Acute inflammation begins in response to a specific injury, leading to the release of soluble mediators, such as cytokines, acute-phase proteins, and chemokines, which promote the migration of neutrophils and macrophages to the site of inflammation to eliminate the trigger or remove damaged cells and initiate healing [2]. However, in chronic inflammatory disorders, this self-limiting response transitions into persistent inflammation, contributing to the development of chronic diseases, including cardiovascular diseases, cancer, autoimmune disorders, and neurodegenerative conditions [3]. Notably, over 60% of global deaths are attributed to chronic inflammatory diseases, underscoring the critical need for effective control of inflammatory responses [4]. Traditionally, steroids and peptides have been utilized for the treatment of inflammatory diseases. However, these therapeutic methods are often accompanied by adverse effects such as an increased risk of infection, osteoporosis, and gastrointestinal disturbances [5,6,7]. Recent studies have highlighted the potential of small molecules as a less harmful alternative for treating inflammatory diseases. These small molecules exhibit promising efficacy while minimizing undesirable side effects. This evolving landscape in anti-inflammatory drug development offers the prospect of more targeted and efficient treatments for inflammatory conditions [8].

The application of machine learning (ML) models has revolutionized drug discovery, offering innovative solutions to long-standing challenges in the field. Among these, Perturbation Theory and Machine Learning (PTML) models have made substantial contributions by integrating diverse chemical and biological data to predict multiple biological outcomes across various targets [9,10,11,12]. This multitarget predictive capability has enabled PTML models to advance therapeutic research in areas such as medicinal chemistry, nanotechnology, oncology, neuroscience, immunology, and infectious diseases. By addressing the limitations inherent to traditional in silico methods, PTML models have paved the way for more comprehensive and accurate predictions [13,14,15]. Furthermore, with the integration of ML tools throughout the drug development process, researchers can accelerate discovery, reduces risks, and improve the overall efficiency of identifying new therapeutic compounds [16].

In this study, we introduce AISMPred, a robust computational approach designed for predicting AISMs. Initially, we developed the model using a comprehensive array of five distinct ML classifiers—RF, ET, KNN, LR, and Ensemble methods—leveraging 2D, FP, and hybrid descriptors. The generated ML models underwent a rigorous 10-fold cross-validation procedure. Subsequently, we ranked these models based on their Area Under the Receiver Operating Characteristic Curve (AUROC) value. Among the 15 ML models, the ET model achieved superior performance, demonstrating higher accuracy and AUC value on the independent test dataset. The proposed AISMPred approach serves as an efficient method for predicting anti-inflammatory small molecules.

## 2. Results and Discussion

To ensure the reliability and effectiveness of our model, we utilized datasets containing 2D, FP, and a hybrid feature set for training and testing purposes. During the model construction process, we applied specialized feature selection techniques tailored to the 2D and hybrid sets. The experimental results produced highly encouraging outcomes, with our model demonstrating exceptional performance during independent testing. These findings underscore our model’s resilience and practical applicability for real-world applications.

### 2.1. Dataset Construction and Feature Selection

The dataset comprising 4300 anti-inflammatory compounds was evenly distributed between AISM and non-AISM instances. The training set comprised 1720 AISM and non-AISM compounds, while the independent test set included 430 AISM and non-AISM compounds. Model development incorporated three key features: 2D descriptors generated by Mordred software (v1.2.0), PubChem fingerprint descriptors computed using PaDel software (v2.21), and a hybrid set combining both sets of descriptors (2D and FP). Acknowledging the varying importance of features, the SVC-L1 regularization method was performed to select the optimum features, identifying 726 descriptors for 2D and 1167 descriptors for the hybrid feature set. No feature selection method was applied to PubChem fingerprints. We employed t-SNE (t-distributed neighbor embedding) [17] to visualize the chemical space of compounds across three feature sets: 726 2D descriptors, 881 fingerprint descriptors, and a hybrid set of 1167 features, as shown in Figure 1. This scatter plot illustrates how reducing high-dimensional data to a two-dimensional representation allowed us to identify distinct clusters of training and test data, reflecting shared chemical properties [18,19]. The overlap between the training and test datasets reveals a shared chemical space, suggesting the model captures consistent patterns that generalize effectively. This shared chemical space is crucial for model validation, as it supports the model’s capacity for unbiased predictions and indicates its potential to perform well on unseen compounds with similar chemical characteristics.

### 2.2. Prediction Analysis of Classifiers Using 2D and FP

The model development process of 2D and FP descriptors involved utilizing a training dataset and implementing various ML algorithms, including RF, ET, KNN, LR, and Ensemble methods. To ensure robustness, a 10-fold cross-validation strategy was applied for evaluating model performance, along with hyperparameter optimization. Firstly, we have developed a model using 2D descriptors. Throughout the training phase, the models displayed diverse accuracies: 0.82 for RF, 0.83 for ET, 0.75 for KNN, 0.77 for LR, and 0.82 for the Ensemble method. Subsequently, their performance was assessed on an independent test dataset, resulting in accuracies of 0.87, 0.90, 0.78, 0.83, and 0.89 for RF, ET, KNN, LR, and Ensemble, respectively. The performance metrics for both training and independent test datasets using 2D descriptors are detailed in Table 1. Notably, RF, ET, and Ensemble models exhibited comparatively similar performance. To determine the optimal model, the AUROC score was employed, revealing values of 0.96, 0.97, and 0.97 for RF, ET, and Ensemble classifiers, respectively, based on the independent test dataset.

Similarly, models were generated using the FP-based descriptors, demonstrating accuracies of 0.83, 0.83, 0.79, 0.78, and 0.83 for RF, ET, KNN, LR, and the Ensemble methods on the training dataset. Further validation with the test set yielded accuracies of 0.90, 0.89, 0.85, 0.89, and 0.89 for RF, ET, KNN, LR, and the Ensemble method, respectively, where once again, RF, ET, and the Ensemble method exhibited superior performance with AUROC values of 0.95, 095 and 0.96. The performance metrics for both training and independent test datasets using FP-based descriptors are detailed in Table 2. Upon analysis of these results from both 2D and FP datasets, ET and the Ensemble method demonstrated superior outcomes.

### 2.3. Prediction Analysis of Classifiers Using Hybrid Features

Utilizing the hybrid feature set, models were developed with accuracies of 0.82, 0.84, 0.74, 0.77, and 0.83 during training and achieved accuracies of 0.88, 0.92, 0.75, 0.85, and 0.91 during testing for RF, ET, KNN, LR, and the Ensemble method, respectively. The optimal parameters for training each ML model are presented in Appendix A. Notably, the ET model exhibited promising results, showcasing superior AUROC values of 0.97 and outperforming others with a Sensitivity (SN) of 0.87, Specificity (SP) of 0.97, and Accuracy (ACC) of 0.92. The performance metrics for training and independent test datasets using hybrid features are detailed in Table 3. The ROC and precision-recall curves of the selected ET model in Figure 2 provide insights into each model’s performance on both the training and independent test sets.

### 2.4. Feature Importance

Feature importance analysis is a technique used to assess the contribution of each input descriptor to model performance [20]. The selected ET model using hybrid features was utilized to rank the features contributing to the prediction, as depicted in Figure 3. Notably, Lipinski, C3SP2, MDEC-23, and other PubChemFP were identified as the top 20 most important features in the predictive model.

The identified molecular features, including MDEC-23, C3SP3, Lipinski’s Rule of Five, and PubChem fingerprints, are essential for understanding the structural and functional characteristics of molecules. MDEC-23 offers valuable information about atomic electronegativity in molecules by considering the effects of adjacent atoms. Lipinski’s Rule of Five evaluates the drug-likeness of a compound, helping to determine its potential for pharmacological or biological activity. C3SP3 describes sp3-hybridized carbons, commonly present in aliphatic chains and saturated rings that provide flexibility for binding interactions, while PubChem fingerprints directly encode functional groups, including saturated or aromatic heteroatom-containing five membered rings and aromatic carbon-only six-membered rings, which are frequently associated with anti-inflammatory agents [21,22]. This comprehensive analysis highlights the collective significance of these features in predicting the anti-inflammatory potential of small molecules, underscoring the value of advanced descriptors in enhancing model accuracy.

## 3. Materials and Methods

### 3.1. AISMPred Computational Architecture

The computational architecture of the ML-based AISM prediction model comprises six key sections, as depicted in Figure 4. (1) Dataset Collection: The initial phase involves the collection of anti-inflammatory small molecules annotated with ${IC}_{50}$ values from the PubChem BioAssay database [23]. (2) Dataset curation and Partitioning: The datasets are meticulously curated and partitioned into distinct training and independent test sets. (3) Feature Calculation: The third step entails feature calculation, a crucial aspect of model development. This process is facilitated by employing the Mordred and PaDel software [24,25]. (4) Feature Selection Method—SVC-L1 Regularization: This is to identify the most relevant features in the dataset to improve the model efficiency and accuracy. (5) Generation of ML Classifiers and Evaluation: Various ML-based models are generated using the selected features and evaluated through a thorough validation process. (6) Performance Metrics Analysis: Evaluation metrics such as sensitivity, specificity, area under the ROC curve (AUC-ROC), and others are computed to measure the predictive capabilities and overall efficacy of the developed AISM prediction model.

### 3.2. Dataset Collection

In this study, a total of 4300 AISMs with annotated ${IC}_{50}$ values were curated using the search query “anti-inflammatory inhibitors” from the PubChem BioAssay database, which is maintained by the National Center for Biotechnology Information (NCBI) (https://www.ncbi.nlm.nih.gov/) (accessed on 5 March 2024). To ensure data quality, we applied refinement steps to remove duplicate molecules and compounds with inconclusive ${IC}_{50}$ values [26,27].

The ${IC}_{50}$ values of the collected compounds were transformed into ${pIC}_{50}$ values to provide a logarithmic scale, which enhances the interpretability and comparability of bioactivity measurements. This transformation facilitates distinguishing between active and inactive compounds, which is essential for accurate classification and screening. A ${pIC}_{50}$ threshold of 6 μM was selected to classify compounds as active or inactive. This cutoff was chosen to be small enough to differentiate active inhibitors from inactive ones while also ensuring a balance that prevents data bias, which could otherwise negatively impact model performance [27]. Compounds with bioactivity values below this threshold ($p{IC}_{50}\;$< 6 μM) were classified as active and those above ($p{IC}_{50}\;$> 6 μM) were classified as inactive.

The final dataset was balanced to include an equal number of active (2150) and inactive (2150) AISMs. This dataset was then split into training and test sets in an 80:20 ratio, with 80% (1720 AISM and non-AISMs) allocated to the training set and the remaining 20% (430 AISM and non-AISMs) to the test set. This balanced dataset facilitated effective classification in predictive modeling, where the training set was used for model development, and the independent test set was reserved for evaluating model performance. The dataset is available through the provided GitHub link in the Data Availability section.

### 3.3. Feature Generation

Molecular descriptors represent the physicochemical properties of molecules and are primarily used in structure–activity relationship (SAR) and quantitative structure–activity relationship (QSAR) studies. These descriptors are critical for constructing QSAR models, which aim to predict the biological activities of unknown compounds based on their structural characteristics [28,29]. In our study, we utilized Mordred software (v1.2.0) [25] and PaDel Software (v2.21) [24] to generate descriptors and fingerprints for training and independent test datasets. The 2D descriptors were calculated using Mordred, a fast, flexible, open-source software that simplifies the calculation of molecular descriptors and offers high customization, making it ideal for large-scale descriptor calculation in modeling. Meanwhile, PubChem fingerprint-based descriptors were generated using PaDel, which excels in calculating different types of fingerprints, surpassing other software in this aspect. Molecules can be characterized by a wide range of properties derived from their structure, including molecular weight, atom counts, bond counts, ring counts, types of carbon atoms, and charge descriptors [30]. A total of 1615 descriptors were calculated for each compound, focusing on 2D properties. The PubChem fingerprints consist of substructure-based fingerprints represented by a vector of 881 bits that were calculated, with each bit encoding the presence of an element or substructure, a specific ring system, atom pairs, and the nearest neighbors of each atom [31].

### 3.4. Feature Selection Approaches

Numerous studies have established that not all features carry equal significance and importance in ML model generation [21,32,33,34]. Therefore, it is crucial to identify the most relevant features from the extensive array of features available. In this study, we employed the SVC-L1 regularization method to select the most relevant features, focusing solely on the 2D and hybrid feature sets [35]. Feature selection was conducted using the L1-penalized Linear Support Vector Classification (LinearSVC) model in conjunction with the SelectFromModel method from the Scikit-learn library [36]. Hyperparameter tuning was performed to select the optimal parameters for identifying the most relevant features during the selection process. Finally, a LinearSVC model was defined with the ‘l1’ penalty, and the dual parameter was set to ‘False’. This penalty promotes sparsity in the learned coefficients, effectively conducting feature selection during model training. L1 regularization imposes a penalty equal to the sum of the absolute values of the coefficients on the loss function, causing some parameters to shrink to zero. As a result, variables associated with these zero-valued coefficients are excluded from the training model, allowing the L1-norm to facilitate the selection of a subset of features. The removal of constant features and normalization were also applied during feature selection to mitigate overfitting and prevent biased results on the test set [37]. The training set was employed for model fitting and hyperparameter optimization through 10-fold cross-validation, while the test set was reserved for model evaluation. By employing the SVC-L1 method, we filtered a total of 726 features for the 2D feature set and 1167 for the hybrid feature set, ensuring that only the most essential features were retained for subsequent analysis.

### 3.5. Machine Learning (ML) Algorithms

In our study, we employed various ML classifiers to develop classification models. The models were constructed using Random forest (RF), ExtraTree (ET) [38], K-Nearest Neighbor (KNN) [39], Logistic regression (LR) [40], and Ensemble method [41]. We utilized the Scikit-learn library package to build these models [36]. RF is an Ensemble learning technique that constructs multiple decision trees during training and generates predictions based on the mode of the classes. ET, a variant of RF, selects random thresholds for each feature and splits nodes using the most optimal threshold among them. KNN, a non-parametric algorithm, assigns a class label to an input sample based on the majority class of its k nearest neighbors in the feature space. LR, a linear classification algorithm, models the probability of binary outcomes using a logistic function, making it suitable for binary classification tasks. The Ensemble method aggregates predictions from multiple base classifiers, enhancing overall performance. In our AISM prediction model, we used a voting classifier technique, where the base classifiers RF and ET independently generated predictions, and the final prediction was determined through a voting mechanism. During the optimization process, we explored a range of hyperparameters to fine-tune the model and avoid overfitting, selecting the most favorable outcomes to generate the final models.

### 3.6. Cross-Validation and Performance Evaluation

The employed approach utilizes 10-fold cross-validation [42] to tune the models for each encoding algorithm, where the 80% training dataset is divided into ten sets. Nine sets are used for training, while the tenth set is reserved for testing. The remaining 20% of the independent test dataset is used for external validation. Model performance is assessed using a comprehensive set of evaluation metrics, including both threshold-dependent (e.g., sensitivity, specificity, accuracy, and MCC) and threshold-independent parameters (e.g., AUROC) [43,44,45,46]. The formulas for the evaluation metrics are shown below.
$$Accuracy=\frac{\left(TP+TN\right)}{\left(TP+TN+FP+FN\right)}$$
$$Sensitivity=\frac{TP}{\left(TP+FN\right)}$$
$$Specificity=\frac{TN}{\left(TN+FP\right)}$$
$$MCC=\frac{\left(TP\times TN)-(FP\times FN\right)}{\surd ((TP+FP)\times (TP+FN)\times (TN+FP)\times (TN+FN))}$$
where TN, TP, FN, and FP state true negative, true positive, false negative, and false positive.

## 4. Conclusions

In this investigation, we have introduced a robust and straightforward predictor designed for the identification of AISM. Recognizing the pivotal role of feature selection, our approach leveraged the SVC-L1 regularization method during the model construction phase. The classification task was carried out, aiming to discern between AISM and non-AISM. The ET model using a hybrid feature set exhibits consistency and stability by achieving superior accuracy of 84% on the training set and 92% on the independent test dataset. Through comprehensive classifier experiments conducted on both the training and an independent test set, our findings underscored the superior performance of the ET classifier in comparison to other models. We anticipate that our proposed method will significantly enhance the accurate prediction of AISM.

## Figures and Tables

**Figure 1 pharmaceuticals-17-01693-f001:**
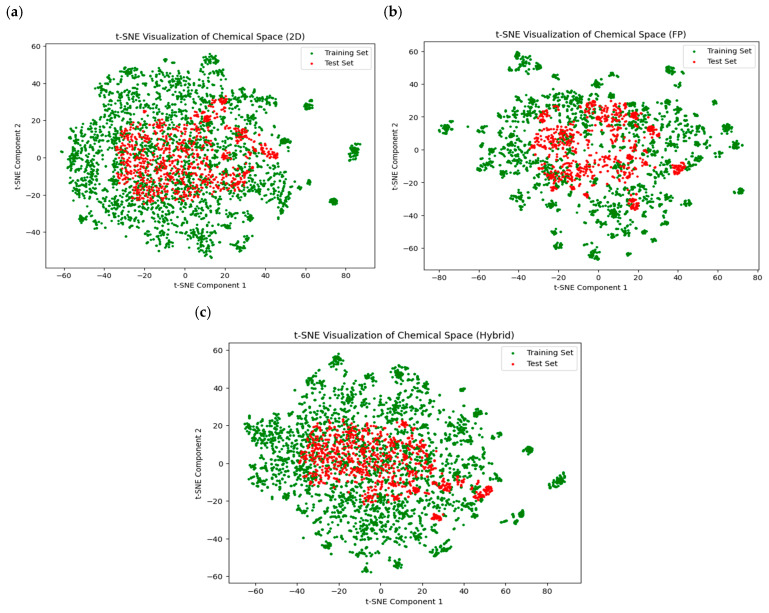
The chemical space of the compounds in the training set compared with that in the test set. (**a**) 2D descriptors, (**b**) fingerprints, (**c**) hybrid (2D + FP).

**Figure 2 pharmaceuticals-17-01693-f002:**
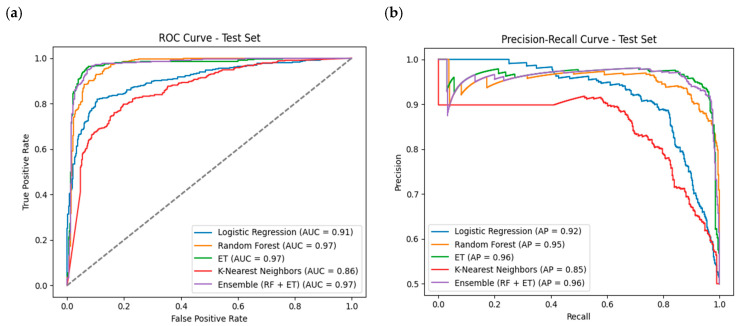
(**a**) Comparison of receiver operating characteristic curves of the four models on external data using Hybrid dataset. The curve closer to the upper left corner showed better overall discrimination ability. (**b**) Comparison of precision-recall curves of the four models on external data. The curve closer to the upper right corner also showed the ability to combine precision with sensitivity. (AP: average precision, AUC: area under the receiver operating characteristic curve, ROC: receiver operating characteristic).

**Figure 3 pharmaceuticals-17-01693-f003:**
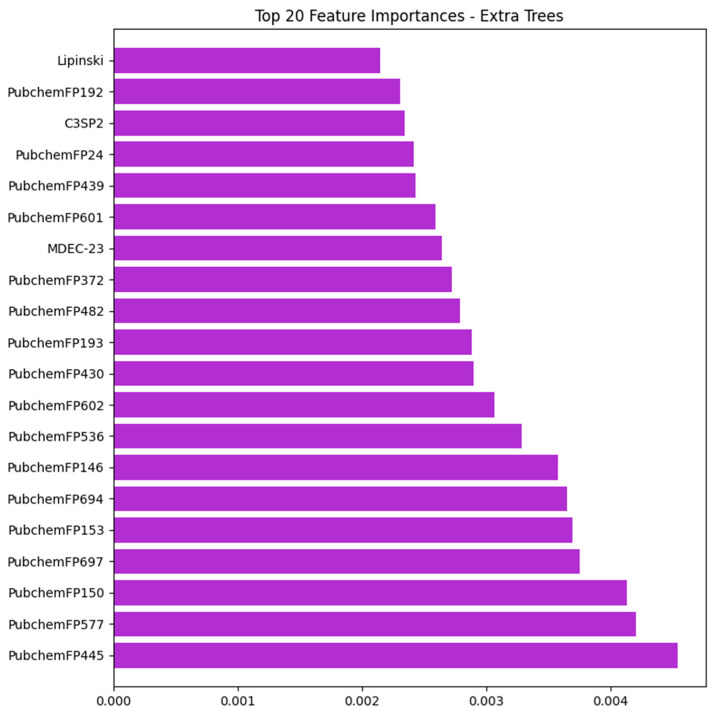
Feature importance plot for the selected ML-based ExtraTree model using hybrid feature set.

**Figure 4 pharmaceuticals-17-01693-f004:**
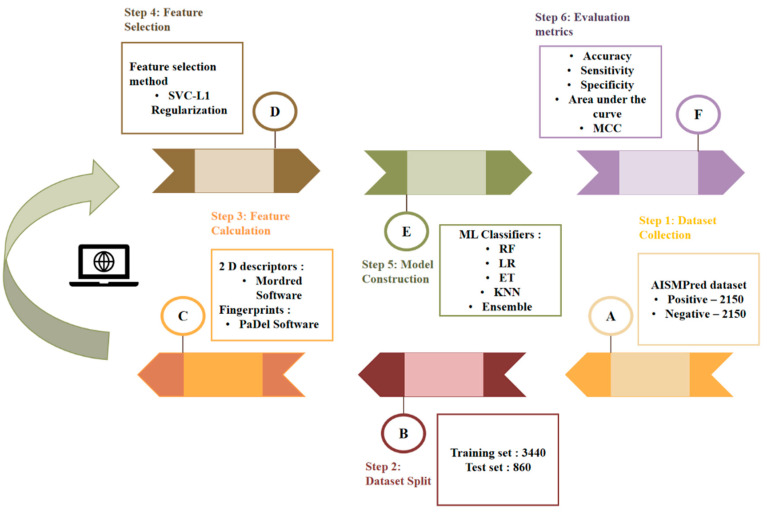
Computational framework of AISMPred. It includes data collection, feature selection, model construction, and performance comparison.

**Table 1 pharmaceuticals-17-01693-t001:** The performance of ML-based models developed using 726 (2D) descriptors.

Algorithms	Training Dataset					Independent Test Dataset				
	AUC	SN	SP	ACC	MCC	AUC	SN	SP	ACC	MCC
RF	0.88	0.83	0.82	0.82	0.65	0.96	0.79	0.95	0.87	0.76
ET	0.88	0.83	0.83	0.83	0.66	0.97	0.85	0.96	0.90	0.81
KNN	0.84	0.85	0.64	0.75	0.51	0.87	0.86	0.70	0.78	0.57
LR	0.83	0.80	0.73	0.77	0.54	0.89	0.80	0.86	0.83	0.67
Ensemble	0.88	0.83	0.82	0.82	0.65	0.97	0.83	0.96	0.89	0.80

Abbreviation: RF—Random forest, ET—ExtraTree, KNN—K-Nearest Neighbor, LR—Logistic Regression, AUC—Area under the curve, SN—Sensitivity, SP—Specificity, ACC—Accuracy, MCC—Mathew’s correlation coefficient.

**Table 2 pharmaceuticals-17-01693-t002:** The performance of ML-based models developed using 881 (FP) descriptors.

Algorithms	Training Dataset					Independent Test Dataset				
	AUC	SN	SP	ACC	MCC	AUC	SN	SP	ACC	MCC
RF	0.88	0.84	0.82	0.83	0.67	0.95	0.84	0.95	0.90	0.90
ET	0.89	0.84	0.83	0.83	0.67	0.95	0.81	0.96	0.89	0.79
KNN	0.86	0.85	0.73	0.79	0.58	0.91	0.90	0.80	0.85	0.71
LR	0.84	0.80	0.75	0.78	0.56	0.95	0.90	0.89	0.89	0.79
Ensemble	0.89	0.83	0.83	0.83	0.67	0.96	0.82	0.96	0.89	0.79

Abbreviation: RF—Random forest, ET—ExtraTree, KNN—K-Nearest Neighbor, LR—Logistic Regression, AUC—Area under the curve, SN—Sensitivity, SP—Specificity, ACC—Accuracy, MCC—Mathew’s correlation coefficient.

**Table 3 pharmaceuticals-17-01693-t003:** The performance of ML-based hybrid models developed after combining 1167 (2D + FP) descriptors.

Algorithms	Training Dataset					Independent Test Dataset				
	AUC	SN	SP	ACC	MCC	AUC	SN	SP	ACC	MCC
RF	0.88	0.83	0.81	0.82	0.65	0.96	0.81	0.95	0.88	0.77
ET	0.89	0.84	0.83	0.84	0.68	0.97	0.87	0.97	0.92	0.85
KNN	0.83	0.85	0.63	0.74	0.50	0.86	0.84	0.66	0.75	0.51
LR	0.82	0.80	0.73	0.77	0.54	0.90	0.81	0.89	0.85	0.70
Ensemble	0.89	0.83	0.83	0.83	0.67	0.97	0.85	0.96	0.91	0.83

Abbreviation: RF—Random forest, ET—ExtraTree, KNN—K-Nearest Neighbor, LR—Logistic Regression, AUC—Area under the curve, SN—Sensitivity, SP—Specificity, ACC—Accuracy, MCC—Mathew’s correlation coefficient.

## Data Availability

All the datasets generated for this study are available at https://github.com/Subathra15/AISMPred/tree/main.

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
