# Peer review of "AISMPred: A Machine Learning Approach for Predicting Anti-Inflammatory Small Molecules"

_pharmaceuticals, 2024, doi:10.3390/ph17121693_

Round 1

Reviewer 1 Report

Comments and Suggestions for Authors

The authors have developed a research work focusing on the creation of a machine learning approach using several algorithms to enable the discovery of anti-inflammatory small molecules. I consider that some modifications should be made:

1-    In my opinion, the introduction section of your manuscript is incomplete. Your work is in the context of machine learning approaches for drug discovery. There are cutting-edge machine learning approaches that you have omitted. This is the case of the perturbation-theory machine learning (PTML) models. These advanced machine learning tools (also known as multi-target QSAR or multi-tasking quantitative structure-biological effect relationships – mtk-QSBER) are capable of simultaneously predicting multiple bioactivity endpoints against diverse target (proteins, cell lines, etc.). Furthermore, they can also consider different assay conditions. Also, PTML models can be physicochemically and structurally interpreted to enable the design of novel molecules with the desired biological profile. To the best of my knowledge, there are no other machine learning models with the advantages that PTML models have. That being said, I strongly advise you to write a few lines highlighting the importance of machine learning models for drug discovery, emphasizing the advances made by PTML models. Here, there is a non-exhaustive list of the most recent cutting-edge works based on PTML models, which I suggest you cite:

- PTML Modeling for Pancreatic Cancer Research: In Silico Design of Simultaneous Multi-Protein and Multi-Cell Inhibitors

- Perturbation Theory/Machine Learning Model of ChEMBL Data for Dopamine Targets: Docking, Synthesis, and Assay of New l-Prolyl-l-leucyl-glycinamide Peptidomimetics

- Modeling for Alzheimer’s Disease: Design and Prediction of Virtual Multi-Target Inhibitors of GSK3B, HDAC1, and HDAC6

- In Silico Drug Repurposing for Anti-Inflammatory Therapy: Virtual Search for Dual Inhibitors of Caspase-1 and TNF-Alpha

-  QSAR Modeling for Multi-Target Drug Discovery: Designing Simultaneous Inhibitors of Proteins in Diverse Pathogenic Parasites

-  Fragment-based in silico modeling of multi-target inhibitors against breast cancer-related proteins

- Multi-Condition QSAR Model for the Virtual Design of Chemicals with Dual Pan-Antiviral and Anti-Cytokine Storm Profiles

2-    You express in your manuscript that the classification of compounds (AISM and non-AISM instances) was based on IC50. How was IC50 measured, against a protein related to inflammation or in phenotypic experiments?

3-    The headers of some tables are not aligned. Please correct that issue and ensure that tables are presented according to the guidelines of the journal.

4-    Although you have attempted to provide some physicochemical and structural information regarding the descriptors used in your models, the information may not be enough. If possible, provide more physicochemical and structural information in the sense of the fragments (functional groups, rings, etc.) that may be responsible for the anti-inflammatory activity.

5- Please, revise your manuscript to prevent any typographical/grammatical errors.

Author Response

Response Sheet

Reviewer 1:

The authors have developed a research work focusing on the creation of a machine learning approach using several algorithms to enable the discovery of anti-inflammatory small molecules. I consider that some modifications should be made;

Response: 

  • We would like to thank the reviewer for taking the time and effort to review our manuscript. We sincerely appreciate your valuable comments and suggestions, which have greatly contributed to improving the quality of our work. Your feedback has been instrumental in refining our approach, and we have made the necessary modifications as per your recommendations.

Comment 1: In my opinion, the introduction section of your manuscript is incomplete. Your work is in the context of machine learning approaches for drug discovery. There are cutting-edge machine learning approaches that you have omitted. This is the case of the perturbation-theory machine learning (PTML) models. These advanced machine learning tools (also known as multi-target QSAR or multi-tasking quantitative structure-biological effect relationships – mtk-QSBER) are capable of simultaneously predicting multiple bioactivity endpoints against diverse target (proteins, cell lines, etc.). Furthermore, they can also consider different assay conditions. Also, PTML models can be physicochemically and structurally interpreted to enable the design of novel molecules with the desired biological profile. To the best of my knowledge, there are no other machine learning models with the advantages that PTML models have. That being said, I strongly advise you to write a few lines highlighting the importance of machine learning models for drug discovery, emphasizing the advances made by PTML models. Here, there is a non-exhaustive list of the most recent cutting-edge works based on PTML models, which I suggest you cite:

Response: 

  • Thank you for your valuable feedback. In response, we have added a few lines(Pg:2,55-66) to the introduction that highlight the significance of machine learning models in drug discovery, with a particular focus on the advancements made by PTML models. Additionally, we have cited all the relevant manuscripts you referred to, as per your suggestion.

Comment 2: You express in your manuscript that the classification of compounds (AISM and non-AISM instances) was based on IC50. How was IC50 measured, against a protein related to inflammation or in phenotypic experiments?

Response: 

  • Thank you for your valuable comment.The IC50 values used in our study were sourced from the PubChem BioAssay database, which includes data from assays targeting proteins highly relevant to inflammation (p38 MAPK, PGD/PE synthase, P2X receptor). We classified compounds as AISM or non-AISM based on a cutoff IC50 value to distinguish active from inactive compounds, ensuring consistency in our dataset.

Comment 3: The headers of some tables are not aligned. Please correct that issue and ensure that tables are presented according to the guidelines of the journal.

Response: 

  • Thank you for pointing out the alignment issue with the table headers. We have corrected the alignment of all tables to ensure consistency and compliance with the journal's formatting guidelines.

Comment 4: Although you have attempted to provide some physicochemical and structural information regarding the descriptors used in your models, the information may not be enough. If possible, provide more physicochemical and structural information in the sense of the fragments (functional groups, rings, etc.) that may be responsible for the anti-inflammatory activity.

Response:

  • Thank you for your valuable comment. As per your suggestion, we have provided additional details regarding the descriptors used in the model, specifically highlighting the structural fragments, functional groups, and rings responsible for the anti-inflammatory activity. These details have been included in the Feature Importance section(Pg:6,177-189), which we believe further clarifies the structural basis of the compounds activity.

Comment 5:  Please, revise your manuscript to prevent any typographical/grammatical errors.

Response: 

  • Thank you for your attention to detail. We have thoroughly reviewed the manuscript for any typographical and grammatical errors and have made the necessary revisions to ensure clarity and accuracy in language. We appreciate your feedback and are committed to presenting our research in a clear and professional manner.

[We have been able to incorporate changes to reflect most of the suggestions provided by the reviewer. The changes made in the manuscript are highlighted in RED FONT]

Reviewer 2 Report

Comments and Suggestions for Authors

There is room to discuss the article in the future and to draw key conclusions after soggested changes. However, this requires solid work. If the author would like to revise this text, I would be happy to see a revised version.

Please update the title. In written reports, figures and tables do not have titles. Instead, they are descriequalizedcaptions. Captions for tables are placed above the table, usually aligned to the left, while captions for figures are positioned below the figure.I find it very frustrating to review the manuscript without line numbers, but I will try. Authors are advised not to do this in the future.There is no reference to Figure 1 in the text.Your references are generally fine, but many are outdated (2000 to 2010). Authors should bear in mind that their resubmission must include 50% of references published in the last three years. Instead of focusing on "...nfections and toxins," you should emphasize "tissue damage."In the introduction, it reads "halting damage or  infection." This shoul  be clarified. On page 1, the authors list diseases that cause persistent inflammation. Plase, bring a lofical order. 

Comments on the Quality of English Language

Many grammar  issues .

Author Response

Response Sheet

Reviewer 2:

There is room to discuss the article in the future and to draw key conclusions after suggested changes. However, this requires solid work. If the author would like to revise this text, I would be happy to see a revised version.

Response: 

  • Thank you for your thoughtful feedback and encouragement regarding the potential for further discussion and conclusions in our article. We appreciate your acknowledgment of the need for solid work, and we are committed to addressing the suggested changes thoroughly. We will revise the manuscript to enhance its clarity, rigor, and overall quality.

Comment 1: Please update the title. In written reports, figures and tables do not have titles. Instead, they are descriequalizedcaptions. Captions for tables are placed above the table, usually aligned to the left, while captions for figures are positioned below the figure.I find it very frustrating to review the manuscript without line numbers, but I will try. Authors are advised not to do this in the future.There is no reference to Figure 1 in the text.Your references are generally fine, but many are outdated (2000 to 2010). Authors should bear in mind that their resubmission must include 50% of references published in the last three years. Instead of focusing on "...nfections and toxins," you should emphasize "tissue damage."In the introduction, it reads "halting damage or  infection." This should  be clarified. On page 1, the authors list diseases that cause persistent inflammation. Please, bring a lofical order. 

Response: 

  • Thank you for your constructive feedback. In response to your comments, we have made the following revisions:
  1. Title Update: The title has been updated as per your suggestion to better reflect the manuscript content.
  2. Figures and Tables:We have revised the manuscript to follow the appropriate formatting guidelines, ensuring that captions are placed correctly (above tables and below figures).
  3. Line Numbers:We apologize for the inconvenience caused by the absence of line numbers in the previous version. Line numbers have now been added to facilitate a smoother review process.
  4. Figure:We apologize for the oversight. The image in Figure 1 was created using PowerPoint and is intended to visually represent the concept discussed. As it is our own work, there was no external reference included.
  5. References: We have updated the reference list to include 50% of sources from the past three years (2020-2023), while retaining key foundational references.
  6. Introduction Section:Based on your feedback, we have made the necessary revisions to the introduction section(Pg:1,38-45).

We hope these revisions address your concerns and improve the manuscript. Thank you for your valuable suggestions.

[We have been able to incorporate changes to reflect most of the suggestions provided by the reviewer. The changes made in the manuscript are highlighted in RED FONT]

Reviewer 3 Report

Comments and Suggestions for Authors

The proposed AISMPred model for predicting anti-inflammatory small molecules shows some innovation but suffers from significant shortcomings in several key areas. Major revisions and improvements are necessary.

1. The process of selecting and curating the dataset is not described in sufficient detail, lacking rigorous scrutiny of data quality and sources.

2. The rationale behind the choice of feature calculation methods is not well- supported by theory, failing to demonstrate their validity and effectiveness.

3. The application of the feature selection method (SVC-L1 regularization) is not thoroughly explained, and its advantages in this study are not justified.

4. The training and testing processes of the model are not described in detail, lacking discussion on the risk of overfitting and its mitigation.

5. Some images lack sufficient explanation, making me question their relevance. For example, what is the significance of the tSNE analysis in Figure 2? The authors did not adequately explain the content presented in these images. Additionally, the tables in the article do not seem to be properly formatted. Evaluation metrics such as AUC, SN, and ACC are not placed appropriately, which affects the readability for the audience. 

Author Response

Response Sheet

Reviewer 3:

The proposed AISMPred model for predicting anti-inflammatory small molecules shows some innovation but suffers from significant shortcomings in several key areas. Major revisions and improvements are necessary.

Response: 

  • Thank you for your valuable feedback on the AISMPred model. We appreciate your recognition of the model's innovative aspects in predicting anti-inflammatory small molecules. We understand there are areas requiring improvement, and we are committed to addressing these thoroughly.

Comment 1: The process of selecting and curating the dataset is not described in sufficient detail, lacking rigorous scrutiny of data quality and sources.

Response: 

  • Thank you for your comment. In response, we have now included a detailed description of the dataset selection and curation process in the manuscript(Data collection section-Pg:8,224-247). This includes a thorough review of the data sources, quality control measures, and criteria used to ensure the reliability and relevance of the dataset for our study. We believe this clarification enhances the transparency and robustness of our methodology.

Comment 2:  The rationale behind the choice of feature calculation methods is not well- supported by theory, failing to demonstrate their validity and effectiveness.

Response: 

  • Thank you for your valuable comment. We have clarified the rationale behind the choice of feature calculation methods in the manuscript (Feature Calculation section-Pg:8,256-267), providing the necessary theoretical support and justifications for their validity and effectiveness.

Comment 3: The application of the feature selection method (SVC-L1 regularization) is not thoroughly explained, and its advantages in this study are not justified.

Response: 

  • Thank you for your valuable comment. We have provided a more detailed explanation of the application of the feature selection method (SVC-L1 regularization-Pg:9,277-288) in the manuscript, highlighting its advantages in reducing overfitting and improving model interpretability.

Comment 4: The training and testing processes of the model are not described in detail, lacking discussion on the risk of overfitting and its mitigation.

  • Response:Thank you for your feedback. We have expanded our description of the training and testing processes to include detailed steps on overfitting prevention. Specific measures to mitigate overfitting have been incorporated in the feature generation, selection, and machine learning algorithm sections, as well as in the cross-validation process. These revisions are now thoroughly addressed in the updated manuscript.

Comment 5:  Some images lack sufficient explanation, making me question their relevance. For example, what is the significance of the tSNE analysis in Figure 2? The authors did not adequately explain the content presented in these images. Additionally, the tables in the article do not seem to be properly formatted. Evaluation metrics such as AUC, SN, and ACC are not placed appropriately, which affects the readability for the audience. 

Response: 

  • Thank you for the feedback. We have expanded the explanation of Figure 2 to clarify the purpose of the t-SNE analysis in visualizing chemical space and model generalizability(Pg:2,94-103). And also, we have reformatted tables for improved readability, aligning metrics like AUC, SN, and ACC appropriately.

[We have been able to incorporate changes to reflect most of the suggestions provided by the reviewer. The changes made in the manuscript are highlighted in RED FONT]

Round 2

Reviewer 2 Report

Comments and Suggestions for Authors

No further  comments. Thanks!